# Entropy Reveals What You Know: An Entropy-Guided Method for Enhancing the Reliability of Large Language Models

## Abstract

While large language models (LLMs) encode vast amounts of knowledge within their parameters for some mainstream entities, factual inconsistencies and untruthfulness in LLMs often lead to unreliable responses and cause significant risks in practical applications. This paper aims to improve model reliability by enhancing consistency in answers to known facts and encouraging refusal to answer for uncertain questions. Specifically, we introduce **SREF**, an entropy-guided approach designed to enhance the reliability of language models by incorporating **S**elf-**REF**erences, models' understanding of rephrasing questions, with inputs. We analyze and reveal the effectiveness of SREF in enhancing model reliability from the perspectives of entropy and KL divergence. Extensive experiments on 12 LLMs demonstrate that outputs generated with SREF yield more reliable results, including an average improvement of 16.01% over the baselines and a 15.10% average improvement in consistency, while also adapting to identify and acknowledge uncertain facts.

## 1 Introduction

Large language models (LLMs) have demonstrated impressive capabilities, attracting significant attention for their performance across various applications (Chowdhery et al., 2023; Mann et al., 2020). These models are being applied to complex tasks such as reasoning, mathematical computation, and planning, positively influencing a range of industries (Olausson et al., 2023; Tonmoy et al., 2024). While LLMs have some reasonable amount of knowledge for some mainstream entities West et al. (2022), a critical challenge remains: LLMs often produce hallucinations, resulting in untruthful and inconsistent responses. This unreliability poses a significant threat, particularly in high-stakes environments like legal, finance, and medicine (Berglund et al., 2024; Sallou et al., 2024; Weidinger et al., 2021). Therefore, it is urgent to develop effective strategies to evaluate and mitigate hallucinations, enhancing the reliability and trustworthiness of LLM responses.

Hallucinations (Wang & Sennrich, 2020; Ji et al., 2023; Huang et al., 2024) in models manifest in several ways, including factual inconsistency, factual fabrication, and faithfulness. This paper focuses on enhancing the factuality and consistency of language model outputs. It aims to ensure that models can accurately express what they know and don't know, while maintaining consistency across their responses. This contributes to the broader goal of developing language models as reliable knowledge bases (Zheng et al., 2024a; Petroni et al., 2019; Wang et al., 2021). Recently, several methods have been proposed, such as scaling models (Liu et al., 2024; Lee et al., 2022), utilizing adversarial training (Penedo et al., 2023), reinforcement learning with human feedback (RLHF) (Ouyang et al., 2022; Wu et al., 2024) and knowledge editing Meng et al. (2022; 2023). However, these methods often entail significant computational overhead and require high-quality annotated data, or they lack general applicability, necessitating model-specific adaptations.

A recent approach to enhancing LLMs involves mimicking human thought processes, allowing these models to self-evaluate and adjust their outputs. (Liang et al., 2024; Kamoi et al., 2024) This method, known as self-correction, aims to improve LLM responses by refining them during inference. Researchers have applied these techniques to enhance model reliability, categorizing them into three types: self-correction with fine-tuning (Ye et al., 2023; Lee et al., 2024), self-correction with exter-

nal information (Gao et al., 2023; Jiang et al., 2023b), and self-correction with prompting (Shinn et al., 2024; Manakul et al., 2023). However, the practicality of intrinsic self-correction remains debated. For instance, Huang et al. (2024) argue that LLMs currently lack the capability for effective self-corrective reasoning, noting that when models are prompted to identify and fix their own errors, their performance can actually decrease.

In line with (Huang et al., 2024), we revisits self-correction methods and finds that these methods depend on advanced inherent capabilities, such as chain-of-thought reasoning or critical evaluation. Models with weaker abilities struggle to improve both consistency and accuracy concurrently, making self-correction challenging. Additionally, these methods often focus solely on performance improvement, neglecting the importance of having LLMs indicate uncertainty when they don't know an answer, which is crucial for ensuring reliability. To address this, we introduce a self-reference method (SREF) that leverages the model's internal knowledge to generate relevant references. The references can reflect the model's understanding of current knowledge and influence the entropy change in the model's responses, thereby enhancing its reliability. It requires only retrieval from the model itself, allowing it to function effectively as a knowledge base without relying on advanced capabilities or external knowledge sources. Additionally, this approach helps the model become aware of what it knows and doesn't know by encouraging it to answer different questions related to the same knowledge, thereby improving both factuality and consistency.

Finally, we tested SREF on 12 models across three datasets and compared it with four prompt-based self-correction methods to validate its effectiveness. We also analyzed SREF's mechanism through entropy and KL divergence, highlighting its advantages in enhancing model reliability.

Our contributions are as follows:

- We propose a self-reference method (SREF) that significantly enhances the accuracy and consistency of factual expression by leveraging the model's own parameterized knowledge, thereby improving its awareness and mastery of information.

- We analyze and reveal SREF's mechanism using entropy and KL divergence, which helps quantify uncertainty and divergence in the model's predictions. This analysis reveals how SREF effectively reduces uncertainty and improves the reliability of large language models.

- SREF consistently outperforms four other self-correction methods across three datasets (NQ, PoPQA, and TriviaQA). It achieves the highest Factuality scores, with an average improvement of 16.01% over the baselines. Additionally, SREF demonstrates a significant consistency improvement of 15.10%, highlighting its superior accuracy and reliability.

## 2 PRELIMINARIES

**Reliable LLMs Definition** This paper focuses on improving the reliability of large language models (LLMs) in terms of Factuality and Consistency. A Reliable LLM should be aware of what it knows and what it does not know, and be able to communicate this uncertainty clearly to the user. Specifically, a reliable LLM should meet the following two criteria: 1) Factuality: The model should accurately convey what it knows and acknowledge when it is uncertain. 2) Consistency: For information the model is certain about, it should provide consistent answers across different variations of the same question.

**Evaluation** Formally, given an LLM $\mathcal{M}$ and a QA dataset $\mathbb{D}$ containing $N$ factoid questions that the LLM should have encountered during training, the model's responses can be classified as correct, unsure, or incorrect. We use two key metrics for evaluating reliability: *Factuality Rate (FR)* measures how often the model provides correct responses for fact-based questions. *Consistency Rate (CR)* measure how consistency for the model's responses. We follow the approach of Zheng et al. (2024b) and generate $m$ distractor options that are similar to the correct answer for each question in $\mathbb{D}$. We then shuffle the answer options to create $m$ versions of the same multiple-choice question, each with a different order, we denote the multiple-choice question dataset as $\mathbb{D}_{mcq}$. Section 4.1 provides the formal definition for each metric.

The use of distractors in measuring consistency provides a robust test of the model's ability to remain stable in its predictions across variations of the same question. Randomizing the order of multiple-

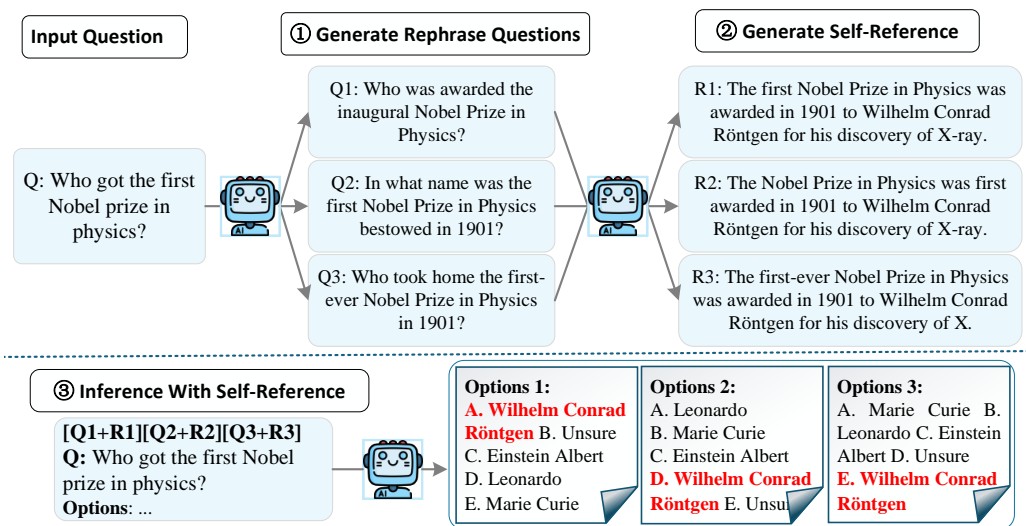

Figure 1: Example of SREF. The model produces more reliable and consistent outputs when combined with self-reference.

choice answers introduces small perturbations that should not affect a reliable model's response if it truly understands the underlying knowledge. Figure 6 provides an example for distractors question.

## 3 GENERATION WITH SELF-REFERENCE(SREF)

The key concept of SREF it evaluates whether the LLM can accurately respond to a question by leveraging its own self-knowledge. For a given input, SREF generates $k$ rephrased versions of the question and simultaneously produces responses to these, which we call self-references. This helps the model assess its mastery of the current knowledge from multiple perspectives. The self-references are then used as an additional prompt alongside the original question to obtain the final result from the model. And helps the model be more confident in its responses or recognize when it lacks sufficient knowledge. SREF is shown in Figure 1. Next, we describe SREF in more detail.

**Rephrase questions generation** Given an input question $x$, the options $o$, a LLM $\mathcal{M}$ and a prompt $x_{pt}$, SREF generates a collection of rephrased questions $\boldsymbol{Q} = \{q_1, q_2, ..., q_k\}$ using $\mathcal{M}$, and each rephrased question expresses the same meaning as the input $x$:

$$\boldsymbol{Q} = \mathcal{M}(x_{pt}||x), \tag{1}$$

where $x_{pt}$ is an instruction for question generation and $||$ denotes concatenation.

**Self-reference generation** Next, SREF uses the same model $\mathcal{M}$ to generate a response $\boldsymbol{R} = \{r_1, r_2, ..., r_k\}$ for each rephrase questions $q_i \in \boldsymbol{Q}$, and we refer to these responses as 'self-references':

$$\boldsymbol{R} = \mathcal{M}(\boldsymbol{Q}). \tag{2}$$

**Inference with self-reference** Finally, SREF concatenate self-references $R$ with the multi-choice question $x$ and the options $o$ then passes this combined input to $\mathcal{M}$.

$$y = \mathcal{M}(p||R||x||o), \tag{3}$$

where $p$ is the prompt for multi-choice question. And $y$ is the response of $\mathcal{M}$. With the influence of $R$, $y$ will be 'Unsure' when $R$ is inconsistency with $x$, see Eq.7 for detail. Figure 7 provides examples of the prompts for different generate tasks.

Self-references intuitively enhance the quality of model responses by ensuring consistency, building coherence, and improving accuracy. For known facts, they provide uniform answers to similar questions, reducing confusion. For uncertain facts, they help express uncertainty, guiding the model to be aware of what it knows and does not.

To verify the role of references, we analyze SREF's mechanism using entropy and Kullback-Leibler (KL) divergence, which help evaluate the consistency and reliability of the model's responses, allowing us to assess how effectively self-references improve the models' performance.

**Self-Reference from the Perspective of Entropy** Self-reference could provides insights into the uncertainty of information and the consistency of model outputs. The key point is that self-reference $R$ could reduce entropy. Since it provides additional context that is already aligned with the model's prior probability distribution, this can act to concentrate probabilities around a few likely tokens, effectively reducing uncertainty. see Theorem 2.6.5 in *Elements of Information Theory*.

For a sequence $X = \{t_1, t_2, ..., t_{n-1}\}$ with $n$ tokens, the joint probability of the sequence generated by the model $\mathcal{M}$ can be expressed as:

$$P(X) = P(t_1)P(t_2|t_1)P(t_3|t_1, t_2)\ldots P(t_n|t_1, t_2, \ldots, t_{n-1}). \tag{4}$$

After we concatenate self-reference $R$ generated by $\mathcal{M}$, the information entropy for the next generated word is:

$$H(t_n|R||X) = -\sum_{t_i} P(t_i|R||X)logP(t_i|R||X). \tag{5}$$

The key question is how entropy changes after concatenating the self-reference $R$ with $X$ and why it provides insights into the uncertainty of LLMs. To answer this, we use Kullback-Leibler (KL) divergence to measure the information lost between between $H(t_n|X)$ and $H(t_n|R||X)$. In this way we could figure out how to use reference to help improve the consistency and reliability.[1]

Let $p(x)$ represent the probability distribution of the input $x$, and $u(x)$ the uniform distribution. The KL divergence is defined as:

$$D_{KL}(p|u) = \sum_x p(x)log(\frac{p(x)}{u(x)}), \tag{6}$$

A lower KL divergence indicates that $p(x)$ is closer to uniform distribution $u(x)$, meaning the possible outcomes have approximately equal probabilities.

After concatenating $R$, the KL is $KL_r = D_{KL}(p(R||xX)|u(X))$, We can then determine:

$$\begin{cases} H(t_n|R||X) \leq H(t_n|X), & KL_r > t, \\ H(t_n|R||X) \geq H(t_n|X), & else. \end{cases} \tag{7}$$

where $t$ is a threshold. This indicates that using $R$ and having a KL divergence larger than $t$ results in more certain model responses, decreasing entropy. And the relationship between reference and question will influence the $KL_r$.

In practice, there are two possible scenarios for varied responses in $R$: 1) different reference express the same meaning, and 2) different reference are inconsistent in expression. In the first scenario, a consistent $R$ is more likely to generate responses relevant to the question $x$. Empirical evidence might show that repeated paraphrasing concentrates the distribution over the next token. This increases model confidence, reinforcing the distribution over the next token $t_n$, and $KL_r$ will increase and make the model more confident. In contrast, in the second scenario, inconsistent references may introduce ambiguity or conflicting information, complicating content generation. This results in a decrease in $KL_r$, make the response more unsure.

Thus, high similarity in references will make the model's response more confidence, and low similarity in references can increase uncertainty in the model's response. Since these references are generated by the model itself, this helps it recognize inconsistencies or gaps in its knowledge.

## 4 EXPERIMENTS

### 4.1 SETUP

**Models and Datasets.**[2] We conduct experiments with publicly accessible model: GPT2-XL(1.5B), GPT-J(6B), LLaMA2-Chat(7B,13B) (Touvron et al., 2023), LLaMA3INSTRUCT(8B),

---

[1]We reference the paper On Information and Sufficiency which shows that Kullback-Leibler (KL) Divergence measures the information lost when using q(x) $q(x)$ to approximate $p(x)$.

[2]We omitted INSTRUCT(Chat) from the model name for short.

| Model | LLaMA2I(7B) | LLaMA2I(13B) | LLaMA3.1I(8B) | GPT4omini | GPT4o | Mean |
|-------|-------------|--------------|---------------|-----------|-------|------|
| **NQ** | | | | | | |
| Vanilla | 30.33% | 24.08% | 63.73% | **69.32%** | **76.28%** | 52.75% |
| RCI | 32.40% | 32.63% | 62.36% | 57.47% | 57.47% | 48.47% |
| SRF | 20.70% | 20.13% | 23.80% | 65.40% | 38.90% | 33.79% |
| SCK | 22.37% | 26.83% | 66.60% | 53.63% | 58.87% | 45.66% |
| SC | 28.57% | 23.50% | 53.67% | 61.03% | 65.27% | 46.41% |
| **SREF** | **51.87%** | **51.39%** | **69.80%** | 61.98% | 67.07% | **60.42%** |
| **PoPQA** | | | | | | |
| Vanilla | 19.61% | 13.48% | **61.55%** | 46.63% | 61.20% | 40.49% |
| RCI | 19.00% | 13.26% | 47.23% | 44.70% | 57.93% | 36.42% |
| SRF | 20.18% | 10.87% | 53.30% | 51.70% | **64.83%** | 40.18% |
| SCK | 20.70% | 16.30% | 51.53% | 38.46% | 62.65% | 37.93% |
| SC | 32.20% | 30.13% | 54.00% | 30.33% | 56.90% | 40.71% |
| **SREF** | **43.82%** | **30.65%** | 60.56% | **51.75%** | 62.82% | **49.92%** |
| **TriviaQA** | | | | | | |
| Vanilla | 34.71% | 33.90% | 64.82% | **72.00%** | **79.00%** | 56.89% |
| RCI | 32.90% | 34.23% | 62.57% | 56.20% | 69.76% | 51.13% |
| SRF | 20.13% | 20.10% | 20.43% | 64.10% | 35.97% | 32.15% |
| SCK | 32.20% | 31.23% | 68.26% | 61.40% | 64.55% | 51.53% |
| SC | 30.90% | 29.10% | 60.83% | 67.03% | 72.80% | 52.13% |
| **SREF** | **50.60%** | **50.46%** | **69.43%** | 68.12% | 74.82% | **62.68%** |

Table 1: Comparison on Micro-FR with different self-correction methods. The best results are in bold, and the second-best results are underlined. "Vanilla" refers to the original model, and "Mean" is the average score across all models. All models used are the instruction version.

LLaMA3.1INSTRUCT(8B,70B), LLaMA3.2INSTRUCT(1B,3B) and MISTRAL-INSTRUCT (7B) (Jiang et al., 2023a). And GPT4 series (Achiam et al., 2023) LLMs: GPT4o-08-06 and GPT4o-mini. For the dataset, We consider three open-domain QA datasets: TriviaQA (Joshi et al., 2017), Natural Questions (NQ) (Kwiatkowski et al., 2019), and PoPQA (Mallen et al., 2023). These datasets are broad-coverage, knowledge-intensive QA datasets, making them well-suited for evaluating LLMs' capacity and consistency to perceive their internal knowledge. Following (Zheng et al., 2024b), during evaluation, we generate $m$ multiple-choice questions for each data instance. We randomly sample 3000 questions for each dataset ($3000 \times m$ multiple-choice questions) for testing. Detailed descriptions of the models and datasets are provided in Appendix A.1.

**Metrics** For factuality, we utilize the Micro-Factuality Correct Rate (Mi-FR) to measure overall accuracy across all multiple-choice questions (MCQs) for a given dataset. The formula is:

$$\text{Micro-FR} = \frac{\sum_{i=1}^{N} \sum_{j=1}^{m} \mathbf{1}_{x_{ij} \text{ is correct}}}{N * m}, \tag{8}$$

where $N$ is the total number of data instances, $m$ is the total number of multi-choices questions for per data, each data contain $\frac{N}{m}$ multi-choices questions. $\mathbf{1}_{x_{ij} \text{ is correct}}$ is an indicator function that equals 1 when Multi-Choices Question $x_{ij} \in \mathbb{D}_{mcq}$ on the j-th Question of the i-th data, and 0 otherwise.

For consistency, we use Macro-Consistency Rate (Macro-CR) to assesses whether all multi-choices questions of a data can yield correct answers.

$$\text{Macro-CR} = \frac{\sum_{i=1}^{N} \mathbf{1}_{T_i=1}}{N}, where \ T_i = \frac{m_{correct}^i}{m} \tag{9}$$

where $m_{correct}^i$ is the number of the correct multi-choice questions for each data. And $\mathbf{1}_{T_i=1}$ is True, meaning all multi-choice answers for the i-th data are correct, indicating a higher consistency for correctness.

**Baselines** We compare SREF with four self-correction methods based on prompting: RCI (Kim et al., 2024), Self-Refine (SRF) (Madaan et al., 2024), Self-check GPT (SCK) (Manakul et al.,

2023), and Self-Consistency (SC) (Wang et al., 2023). All of these methods refine LLMs' initial responses solely by leveraging the models' inherent capabilities, without relying on external feedback or supervised fine-tuning. Detailed descriptions of the baselines are provided in Appendix A.2.

## 4.2 RESULTS

### 4.3 RESULTS ON FACTUALITY RATE AND CONSISTENCY RATE

Table 1 presents the Micro-FR results comparing self-correction based on prompting. A higher Micro-FR indicates higher accuracy. And SREF achieves the highest Micro-FR scores for LLaMA2 (7B) and (13B) across multiple datasets, and also obtains the highest Mean score across all models.

**SREF can significantly improve the factuality of models.** SREF consistently delivers the best or second-best Micro-FR across various models and datasets, showcasing its strong ability to enhance accuracy in knowledge-intensive tasks. Unlike methods such as RCI and REFINE, which depend on the model's own feedback loops and self-correction, SREF enables models to retrieve relevant information from itself, significantly boosting both factuality. This makes it particularly effective in addressing the limitations of models lacking high-level inherent capabilities. For smaller-scale models like LLaMA2(7B) and LLaMA2(13B), SREF produces dramatic improvements, highlighting its scalability and robustness in enhancing both factuality and consistency.

| | LLaMA2I(7B) | LLaMA2I(13B) | LLaMA3.1I(8B) | GPT4omini | GPT4o | Mean |
|---|---|---|---|---|---|---|
| **NQ** | | | | | | |
| Vanilla | 5.32% | 3.59% | 22.57% | **57.76%** | **67.20%** | 31.29% |
| RCI | 10.80% | **14.60%** | 24.20% | 50.20% | 50.20% | 30.00% |
| SRF | 3.00% | 3.10% | 7.80% | 46.80% | 14.40% | 15.02% |
| SCK | 4.80% | 12.60% | **42.40%** | 49.40% | 52.40% | 32.32% |
| SC | 4.40% | 2.20% | 20.00% | 50.60% | 58.20% | 27.08% |
| **SREF** | **13.94%** | 13.58% | 27.00% | 54.43% | 59.70% | **33.73%** |
| **PoPQA** | | | | | | |
| Vanilla | 3.02% | 1.47% | 22.48% | 34.33% | 49.90% | 22.24% |
| RCI | 4.20% | 3.60% | 21.40% | 40.00% | 49.20% | 23.68% |
| SRF | 2.97% | 2.30% | 24.60% | 39.00% | 51.60% | 24.09% |
| SCK | 4.40% | 5.40% | **35.80%** | 36.60% | 56.98% | 27.84% |
| SC | 9.80% | **8.20%** | 20.00% | 34.60% | 48.80% | 24.28% |
| **SREF** | **12.01%** | 7.50% | 21.94% | **46.10%** | 57.30% | **28.97%** |
| **TriviaQA** | | | | | | |
| Vanilla | 6.65% | 6.92% | 24.36% | **62.36%** | 71.00% | 34.26% |
| RCI | **14.60%** | 12.80% | 22.00% | 44.20% | 58.60% | 30.44% |
| SRF | 4.25% | 3.51% | 5.00% | 45.40% | 11.80% | 13.99% |
| SCK | 11.40% | **16.80%** | **40.00%** | 54.00% | 53.17% | 35.07% |
| SC | 5.00% | 4.00% | 19.00% | 57.00% | 63.60% | 29.72% |
| **SREF** | 13.14% | 13.51% | 26.75% | 58.47% | **71.90%** | **36.75%** |

Table 2: Comparison on Macro-CR with different self-correction methods.

Table 2 shows the results on Macro-CR, where a higher Macro-CR indicates that the model can provide consistent and correct responses.

**SREF can stability improve the consistency of LLMs, and also enhances the capabilities of GPT4o and GPT4omini, particularly in improving consistency.** For LLMs such as GPT4o and GPT4omini, they generally lead in performance, especially in Vanilla settings, showcasing their strong baseline capabilities and robustness across different datasets. While SC and SCK methods also contribute to improvements—with SC enhancing overall accuracy and SCK improving consistency—neither matches the stable impact of SREF. Furthermore, SREF provides significant Macro-CR improvements, achieving a top score of 46.10% for GPT4omini in PoPQA and 71.90% for GPT4o in TriviaQA. It also obtains the highest Mean score across all models. These results clearly demonstrate SREF's superior ability to enhance both accuracy and consistency, making it a key method for optimizing model performance and guiding future development strategies.

Notably, when dealing with models that have a large number of parameters or are more powerful, such as GPT-4o and LLaMA 3.1, other baselines may exhibit advantages on certain datasets. This is related to their reliance on the model's capabilities. However, when the Vanilla model's capabilities are weaker, these methods struggle to be effective. In contrast, SREF relies solely on the

---

**When was the last time michigan beat ohio state? (Type: False → Unsure)**

Reference 1: *When was the last time michigan beat ohio state? A:in football? The last time Michigan beat Ohio State in football was on November 18, 2018, when the ...*
Reference 2: *When was the last time Michigan beat Ohio State in a football game at the Big House? Answer: The last time Michigan beat Ohio State in a football game at the Big House was on November 18, 2003 ...*
Reference 3: *When did the Wolverines last emerge victorious over the Buckeyes in a regular season matchup? A: That would be in 2003, when Michigan won 35-21 in Columbus ...*
*Options:* A. 2002 C. 2018 D. Unsure E. 2011 F. 2012 G. 2003
*Reference score: 0.3*
**SREF: C. 2018 → D. Unsure**
**SC: (Wang et al., 2023) C. 2018 → G. 2003**

---

Figure 2: An example of False to Unsure with Reference.

model's ability to represent knowledge, allowing it to consistently improve overall performance and be suitable for Vanilla models of varying scales.

### 4.4 RESULTS FOR UNCERTAINTY

Table 4 reports the performance regarding uncertainty. From the table, we can see that our method increases the likelihood of models outputting 'unsure' across most models and datasets. Our conclusions align with Zheng et al. (2024a); Zhou et al. (2024), that models like LLaMA3 exhibit some tendencies toward excessive introspection. Therefore, enhancing their ability to respond with 'unsure' helps in building more trustworthy LLMs. For relatively smaller models like LLaMA2, reducing 'unsure' responses suggests an improvement in the information provided by the model. In summary, our method can adaptively help models recognize what they know and don't know based on the reference.

Additionally, we provide a case illustrating how our method helps the model recognize its own uncertainty. As shown in Figure 2, when the model generates references unrelated to the question but maintains consistency among them, the SC method tends to choose the most frequent answer, leading to incorrect responses. In contrast, our method, SREF, detects the LLM's uncertainty from these references and selects 'Unsure.' Further analysis in Section 4.5.1 confirms that choosing 'Unsure' is both necessary and accurate.

### 4.5 ANALYSIS

The above results indicate that SREF can enhance the model's factuality and consistency, and could enhance the ability of aware what LLMs known and don't. In this section, we perform additional experiments to analyze the importance of self-reference.

#### 4.5.1 ANALYZING SREF FROM THE PERSPECTIVES OF ENTROPY AND KL DIVERGENCE

As shown in Figure 3, the confidence score is determined by using models like GPT-4o to assess the consistency of references with the questions. The entropy is computed from the distribution of tokens predicted by the model. Each node in the figure is split into two colors: the left side represents the result of the model's original output, while the right side shows the result with SREF.

Figure 4 illustrates the relationship between KL divergence and the consistency score. The KL divergence is calculated using the SREF distribution and a uniform distribution (see Equation 6). Section A.3 provides more detailed settings and case study.

**The references generated by SREF can adaptively adjust the changes in output entropy and KL divergence, thereby enhancing the model's confidence in known knowledge and its awareness of unknown knowledge.** Entropy measures the model's uncertainty in its predictions, with lower entropy indicating greater confidence. KL divergence quantifies the change in the prediction distribution when incorporating references, with lower KL divergence indicating less confidence.

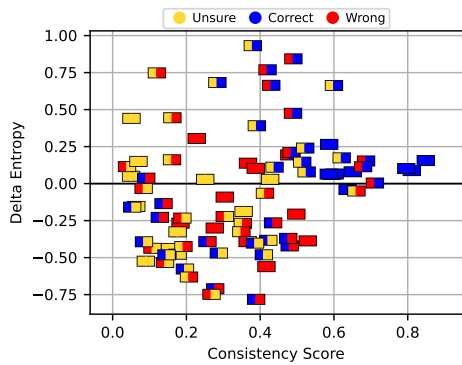 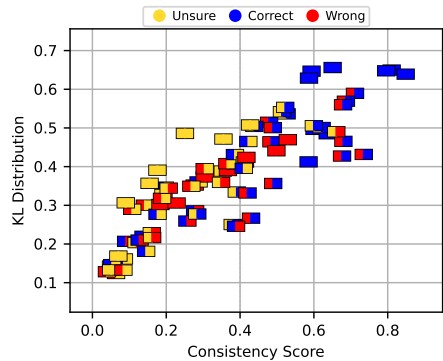

Figure 3: Distribution of Delta Entropy          Figure 4: Distribution of KL divergence

*In terms of entropy*, as seen in Figure 3, the concentration of blue nodes (Correct) in the upper right corner indicates that when our method corrects previously incorrect data, it decreases entropy, enhancing the model's accuracy and confidence. Conversely, the clustering of yellow nodes (Unsure) in the lower left corner suggests that low relevance in generated references increases entropy, signaling uncertainty to the model.

*Regarding KL divergence*, Figure 4 shows that a high consistency score with large KL divergence corresponds to increased certainty, as indicated by the spread of blue and red nodes (Correct and Wrong). In contrast, a low consistency score with small KL divergence, associated with yellow nodes (Unsure), reflects low certainty and a tendency towards uncertainty.

**Overall, through the analysis of entropy and KL divergence, we find that when the references generated by SREF are consistent with the question, the model tends to give a confident and correct response. Conversely, when the consistency score is low, it helps the model recognize its uncertainties.**

### 4.6 DISCUSSION

#### 4.6.1 ANALYZING LLMS RESPONSE CHANGES THROUGH THE CONSISTENCY SCORES

We analyzed the changes in model output when references with different scores were provided. Figure 5 shows the results for models smaller than 13B, and Figure 5c and Figure 5b provides detailed results for GPT4o and GPT4omini across three datasets. The percentage indicates the proportion of the current type, and the number represents the average consistency score of references for the current type.

**SREF can activate the correct knowledge in the model thus improving accuracy when generate high-consistency references. Meanwhile, low-consistency references reflect the model's uncertainty about current knowledge, leading it to choose unsure.**

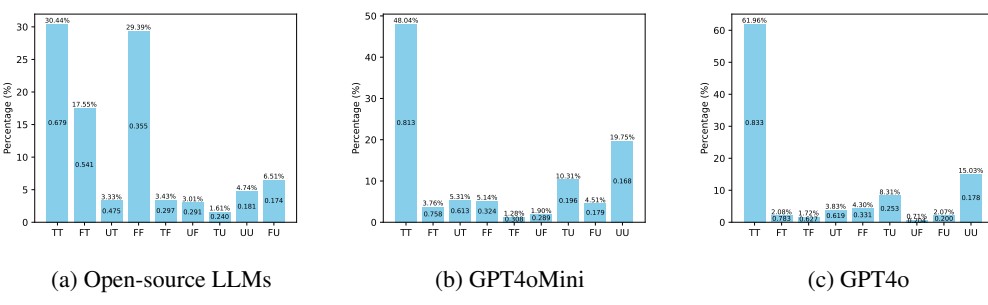

(a) Open-source LLMs          (b) GPT4oMini          (c) GPT4o

Figure 5: Distribution of Response Change across Different Models

As shown in the figure 5, two results show a trend: when the consistency score is high, TT, UT, and FT transitions are most common, indicating that higher scores are more likely to improve model performance. Conversely, when the score is low, the model is more likely to respond with uncertainty. This demonstrates that when references are unrelated to the question, the model becomes unsure about the knowledge related to the question, leading to a preference for uncertain responses.

### 4.6.2 How important are the number of self-references?

To assess the impact of varying numbers of references on the model's performance, we compared results using 2, 3, and 4 references, as shown in Table 6. The table indicates that using 4 references yields the best results, with a notable improvement in accuracy. Fewer references may not adequately capture the model's understanding of the current knowledge, while more references increase computational costs and affect efficiency. In this paper, we simply generate rephrased questions to create the references. The design of these questions is crucial, as well-crafted questions may enhance the relevance and quality of the references. Future research could explore optimizing both the number of references and the design of questions to improve performance without significantly increasing computational demands.

## 5 Related Wrok

### 5.1 Large Language Model as Reliability Knowledge Base

Large language models (LLMs) are powerful knowledge bases, capable of storing and retrieving vast information. However, they often struggle with consistency and accuracy. Foundational work by Petroni et al. (2019) and Roberts et al. (2020) explored their potential and limitations. While efforts like those by Wang et al. (2021) have enhanced factual memorization, reasoning challenges persist (He et al., 2024). To address these, Zheng et al. (2024a) and Zheng et al. (2024b) proposed criteria for ensuring factuality and consistency. At the same time, Jang et al. (2022) and Ribeiro et al. (2019) identified gaps in output stability, and Cohen et al. (2023) and Xue et al. (2023) tackled factual inaccuracies, emphasizing the need for models to recognize their knowledge limits (Cheng et al., 2024; Chen et al., 2024). However, many existing solutions require significant resources (Amayuelas et al., 2023). This paper introduces SREF, a self-referencing framework that enhances LLM reliability. By improving consistency and accuracy, SREF allows models to manage uncertainty more effectively, increasing their trustworthiness as knowledge bases and offering a more efficient solution compared to existing methods.

### 5.2 Self-correction Methods for LLMs

Self-correction methods for large language models (LLMs) enhance reliability through iterative refinement and error correction. Madaan et al. (2024) introduced Self-refine, leveraging self-feedback to improve quality, while Welleck et al. (2023) focused on self-correcting during sequence generation. Pan et al. (2024) surveyed diverse correction strategies, highlighting the need for robust error identification systems. Gao et al. (2023) and Yu et al. (2023) explored self-review and retrieval feedback, respectively, to enhance performance. Additionally, Kim et al. (2024) and Shinn et al. (2024) discussed feedback for task-solving and reinforcement learning. Jung et al. (2022) examined logically consistent reasoning. Despite these advancements, Huang et al. (2024) and Stechly et al. (2023) noted limitations in reasoning, where models struggle to recognize errors independently. In contrast, SREF uses self-referencing to improve consistency and accuracy, addressing these limitations by enhancing inherent reliability.

## 6 Conclusion

In this paper, we introduce SREF, an entropy-guided approach that enhances the reliability of large language models (LLMs) by incorporating self-references. We evaluated its effectiveness through theoretical analysis and experimental validation, showing that self-references can improve accuracy and help models recognize their knowledge boundaries. SREF was tested on three datasets and 12 models, demonstrating its ability to enhance reliability and assess uncertain knowledge. These results highlight SREF's potential to increase the trustworthiness of LLMs in practical applications.

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

# A  APPENDIX

> **Q: Who got the first Nobel prize in physics?**
>
> *Options 1:*
> **A. Wilhelm Conrad Röntgen** B. Unsure C. Einstein Albert D. Leonardo E. Marie Curie
> *Options 2:*
> A. Leonardo B. Marie Curie C. Einstein Albert **D. Wilhelm Conrad Röntgen** E. Unsure
> ...
> *Options $m$:*
> A. Marie Curie B. Leonardo C. Einstein Albert D. Unsure **E. Wilhelm Conrad Röntgen**

Figure 6: An example of Multiple-Choice Question with different option orders.

## A.1  IMPLEMENTATION DETAILS

**Datsete** Natural Questions (NQ) Kwiatkowski et al. (2019) includes questions sourced from web queries, each paired with a corresponding Wikipedia article containing the answer. TriviaQA Joshi et al. (2017) comprises questions from Quiz League websites, supplemented by web pages and Wikipedia searches that may contain the answer. PopQA Mallen et al. (2023) targets long-tail entities. The dataset uses the Wikipedia dump from December 2018 in its retrieval-augmented baseline, indicating that the knowledge in PopQA is covered by this Wikipedia version.

We sampled 3,000 data points from each of these three datasets as evaluation data and generated corresponding multiple-choice questions based on them. Figure 6 give an example for the multiple-choice question.

| Dataset | Model | Vanilla | SREF | Vanilla | SREF | Vanilla | SREF |
|---|---|---|---|---|---|---|---|
| | | Micro-FR | | Macro-CR | | Macro-FU | |
| NQ | LLaMA3.2(1B) | 46.47% | 47.78% | 9.94% | 12.08% | 5.77% | 11.21% |
| | GPT2XL(1.5B) | 15.12% | 15.10% | 0.00% | 0.00% | 13.92% | 11.13% |
| | LLaMA3.2(3B) | 56.81% | 60.74% | 18.04% | 21.12% | 8.77% | 14.64% |
| | GPTJ(6B) | 16.02% | 19.22% | 0.00% | 0.03% | 10.80% | 5.22% |
| | MISTRAL(7B) | 57.12% | 60.76% | 18.26% | 21.10% | 1.74% | 2.05% |
| | LLaMA2(7B) | 30.33% | 51.87% | 5.32% | 13.94% | 19.97% | 17.07% |
| | LLaMA3(8B) | 64.83% | 68.48% | 23.66% | 25.83% | 1.35% | 4.61% |
| | LLaMA3.1(8B) | 63.73% | 69.80% | 22.57% | 27.00% | 2.93% | 4.84% |
| | LLaMA2(13B) | 24.08% | 51.39% | 3.59% | 13.58% | 0.67% | 9.64% |
| | LLaMA3.1(70B) | 70.54% | 73.33% | 28.69% | 31.03% | 2.69% | 3.55% |
| | GPT4omini | 69.32% | 61.98% | 57.76% | 54.43% | 14.64% | 27.90% |
| | GPT4o | 76.28% | 67.07% | 67.20% | 59.70% | 13.45% | 22.32% |
| POPQA | LLaMA3.2(1B) | 39.12% | 37.17% | 7.29% | 8.86% | 14.68% | 27.71% |
| | GPT2XL(1.5B) | 13.87% | 15.30% | 0.00% | 0.00% | 14.13% | 9.42% |
| | LLaMA3.2(3B) | 47.87% | 39.94% | 14.57% | 12.29% | 19.12% | 33.58% |
| | GPTJ(6B) | 14.37% | 16.93% | 0.03% | 0.03% | 13.27% | 4.49% |
| | MISTRAL(7B) | 52.07% | 55.33% | 15.74% | 18.49% | 1.67% | 0.99% |
| | LLaMA2(7B) | 19.61% | 43.82% | 3.02% | 12.01% | 21.02% | 21.29% |
| | LLaMA3(8B) | 60.14% | 60.36% | 22.08% | 22.62% | 2.73% | 11.32% |
| | LLaMA3.1(8B) | 61.55% | 60.56% | 22.48% | 21.94% | 4.11% | 14.95% |
| | LLaMA2(13B) | 13.48% | 30.65% | 1.47% | 7.50% | 0.13% | 10.37% |
| | LLaMA3.1(70B) | 59.22% | 59.21% | 21.11% | 24.17% | 1.91% | 10.24% |
| | GPT4omini | 46.63% | 51.75% | 34.33% | 46.10% | 34.80% | 38.20% |
| | GPT4o | 61.20% | 62.82% | 49.90% | 57.30% | 26.93% | 31.39% |
| TriviaQA | LLaMA3.2(1B) | 41.76% | 45.92% | 8.75% | 11.00% | 8.96% | 15.28% |
| | GPT2XL(1.5B) | 14.24% | 15.28% | 0.00% | 0.00% | 14.21% | 10.92% |
| | LLaMA3.2(3B) | 54.72% | 56.74% | 18.43% | 18.95% | 11.81% | 17.85% |
| | GPTJ(6B) | 15.71% | 19.43% | 0.00% | 0.07% | 11.09% | 6.50% |
| | MISTRAL(7B) | 60.47% | 65.45% | 20.94% | 23.25% | 1.59% | 1.08% |
| | LLaMA2(7B) | 34.71% | 50.60% | 6.65% | 13.14% | 31.53% | 18.38% |
| | LLaMA3(8B) | 64.29% | 66.37% | 24.16% | 24.76% | 3.01% | 6.48% |
| | LLaMA3.1(8B) | 64.82% | 69.43% | 24.36% | 26.75% | 4.13% | 6.36% |
| | LLaMA2(13B) | 33.90% | 50.46% | 6.92% | 13.51% | 3.92% | 9.82% |
| | LLaMA3.1(70B) | 73.18% | 76.18% | 31.06% | 32.97% | 2.55% | 2.78% |
| | GPT4omini | 72.00% | 68.12% | 62.36% | 58.47% | 16.08% | 26.60% |
| | GPT4o | 79.00% | 74.82% | 71.00% | 71.90% | 12.53% | 21.27% |

Table 3: Results on three dataset across 12 models.

## A.2 MODEL AND BASELINES

**Models** For all models, we used the weights provided by Hugging Face. The results for all models are shown in Table 3.

**Baselines** We compare SREF with four self-correction methods based on prompting: RCI (Kim et al., 2024), Self-Refine (SRF) (Madaan et al., 2024), Self-check GPT (SCK) (Manakul et al., 2023), and Self-Consistency (SC) (Wang et al., 2023). All of these methods refine LLMs' initial responses solely by leveraging the models' inherent capabilities, without relying on external feedback or supervised fine-tuning. The prompts for different baselines are shown in Figure 8

Recursive Criticism and Improvement (RCI) Kim et al. (2024) focuses on enhancing language models' reasoning capabilities by structuring problems as a sequence of intentions or intermediate steps. This approach aims to improve logical reasoning and decision-making processes in complex tasks. RCI relies on the model's ability to generate explanations and evaluations of its own responses and refine them based on these evaluations. However, this method is less effective in models that lack chain-of-thought capabilities and strong feedback mechanisms. The prompt we used in RCI is:

Self-Refine (SRF) Madaan et al. (2024)is a method where models iteratively refine their outputs by self-evaluating and making adjustments. This technique allows models to improve the quality and accuracy of their responses through multiple iterations. However, it requires the model to recognize inaccuracies in its responses, which can be challenging for smaller-scale models.

Self-check GPT (SCK) (Manakul et al., 2023) involves a model verifying its own outputs by generating explanations or justifications. This self-checking mechanism helps ensure the reliability and correctness of the model's responses. Self-Consistency (SC) (Wang et al., 2023) enhances model performance by generating multiple outputs for a given input and selecting the most consistent re-

sult. This method leverages the diversity of responses to arrive at a more reliable and robust answer. These two papers are related to our work. SCK focuses on evaluating hallucinations in language models, with the model determining whether the evidence supports its response. In our setting, we use these responses and feedback as references to enhance the model's output. For Self-Consistency, it involves sampling multiple times to generate diverse responses to improve the model, but it relies heavily on sampling methods. In tasks involving short question answers, it may lack consistency because different samples often yield the same response.

|  | LLaMA2I(7B) | LLaMA3.1I(8B) | LLaMA2I(13B) | GPT4omini | GPT4o |
|---|---|---|---|---|---|
| | | | NQ | | |
| Vanilla | 19.97% | 2.93% | 0.67% | 14.64% | 13.45% |
| SREF | 17.07% | 4.84% | 9.64% | 27.90% | 22.32% |
| | | | PopQA | | |
| Vanilla | 21.02% | 4.11% | 0.13% | 34.80% | 26.93% |
| SREF | 21.29% | 14.95% | 10.37% | 38.20% | 31.39% |
| | | | TriviaQA | | |
| Vanilla | 31.53% | 4.13% | 3.92% | 16.08% | 12.53% |
| SREF | 18.38% | 6.36% | 9.82% | 26.60% | 21.27% |

Table 4: Results for Uncertainty. The value represents the ratio of the total number of 'unsure' responses to the total number of questions.

## A.3 DETAILS OF ANALYSIS

### A.3.1 RESULTS FOR UNCERTAINTY.

Table 4 evaluates the effectiveness of various language models—LLaMA2I (7B), LLaMA3.1I (8B), LLaMA2I (13B), GPT4omini, and GPT4o—across three datasets: NQ, PopQA, and TriviaQA, focusing on the percentage of "unsure" responses. The Vanilla method generally results in lower "unsure" response rates, while the SREF method significantly increases these rates, indicating improved uncertainty recognition. Notably, models like LLaMA2I (13B) and GPT4omini show substantial increases in "unsure" responses with SREF, especially in the NQ and PopQA datasets. This suggests that SREF effectively enhances the models' ability to identify uncertainty, leading to more cautious and reliable outputs. The impact of SREF varies by model and dataset, highlighting differences in how each model handles uncertainty and the influence of dataset characteristics on model performance.

### A.3.2 DETAILS OF CONSISTENCY SCORE (CS) AND DELTA ENTROPY (DE)

To provide insights into the SREF effectiveness, we evaluate on LLaMA2 (7B) and LLaMA3.1 (8B) across datasets Natural Questions (NQ), TriviaQA, and PoPQA. We show the results before and after applying SREF in Table 5, categorizing them as True to True (TT), False to True (FT), True to False (TF), Unsure to True (UT), Unsure to Unsure (UU), and Mixed Types (FU, UF). We also provide the transitions of GPT4o and GPT4omini in Figure 5b and 5c. UT transitions, where models move from uncertain to correct, show that references can clarify and enhance confidence. TU and FU transitions, where models become uncertain, suggest that certain references can make the model aware what they know and don't know. An outlier is the TF transition in Figures 5b and 5c, where the consistency score is high, but the result worsens. This suggests that larger models such as GPT4o are susceptible to being influenced, even when provided with factually relevant references.

## A.4 ALBATION STUDY

To test the impact of different numbers of references on the model's performance, we compared the results when providing 2, 3, and 4 references, as shown in Table 6. The table indicates that using 4 references yields the best results. Fewer references may not adequately reflect the model's understanding of the current knowledge, while more references increase computational costs and affect efficiency.

Overall, LLaMA3.1 (8B) outperforms LLaMA2 (7B) across all datasets and metrics. Increasing the number of references generally enhances Mi-FR and Ma-CR, indicating improved factual accuracy

| Model | Type | CS | DE | ND | Type | CS | DE | ND | Type | CS | DE | ND |
|---|---|---|---|---|---|---|---|---|---|---|---|---|
| | | | NQ | | | | PoPQA | | | | TriviaQA | |
| LLaMA2(7b) | TT | 0.594 | 0.062 | 4656 | TT | 0.587 | 0.265 | 2111 | TT | 0.651 | 0.080 | 5009 |
| | TF | 0.278 | -0.710 | 552 | TF | 0.257 | -0.392 | 214 | TF | 0.389 | -0.782 | 751 |
| | TU | 0.196 | -0.577 | 274 | TU | 0.176 | -0.477 | 87 | TU | 0.288 | -0.469 | 333 |
| | FT | 0.479 | 0.194 | 3502 | FT | 0.485 | 0.210 | 3643 | FT | 0.489 | 0.475 | 1903 |
| | FF | 0.277 | -0.300 | 4107 | FF | 0.184 | -0.267 | 4340 | FF | 0.304 | -0.091 | 3427 |
| | FU | 0.142 | -0.533 | 1329 | FU | 0.108 | -0.437 | 2129 | FU | 0.189 | -0.233 | 622 |
| | UT | 0.526 | 0.079 | 1190 | UT | 0.439 | 0.111 | 1516 | UT | 0.514 | 0.145 | 1832 |
| | UF | 0.353 | -0.268 | 938 | UF | 0.192 | -0.425 | 1592 | UF | 0.349 | -0.325 | 1571 |
| | UU | 0.155 | -0.438 | 1446 | UU | 0.090 | -0.524 | 2362 | UU | 0.176 | -0.326 | 2546 |
| LLaMA3.1(7B) | TT | 0.602 | 0.065 | 10633 | TT | 0.586 | 0.064 | 7446 | TT | 0.637 | -0.040 | 10478 |
| | TF | 0.138 | -0.135 | 582 | TF | 0.091 | 0.037 | 1172 | TF | 0.128 | -0.228 | 595 |
| | TU | 0.085 | -0.393 | 284 | TU | 0.053 | -0.159 | 794 | TU | 0.146 | -0.481 | 351 |
| | FT | 0.490 | 0.843 | 1706 | FT | 0.430 | 0.664 | 1878 | FT | 0.420 | 0.769 | 1561 |
| | FF | 0.367 | 0.138 | 3862 | FF | 0.228 | 0.305 | 4731 | FF | 0.388 | 0.103 | 3667 |
| | FU | 0.064 | -0.154 | 400 | FU | 0.039 | 0.115 | 1236 | FU | 0.084 | -0.033 | 483 |
| | UT | 0.391 | 0.835 | 211 | UT | 0.380 | 0.932 | 179 | UT | 0.284 | 0.683 | 305 |
| | UF | 0.161 | 0.539 | 129 | UF | 0.120 | 0.748 | 227 | UF | 0.162 | 0.445 | 199 |
| | UU | 0.050 | 0.202 | 187 | UU | 0.051 | 0.441 | 331 | UU | 0.070 | 0.150 | 355 |

Table 5: Details of Consistency Score (CS) and Delta Entropy (DE), Number of Data means the number of each types data.

and consistency. However, the highest CR isn't always achieved with the maximum number of references, suggesting a balance between reference quantity and quality. Specifically, LLaMA3.1 (8B) shows significant gains in the PoPQA dataset, with Mi-FR and CR peaking at 60.56% and 82.77% with four references. This analysis underscores the effectiveness of using multiple references in SREF to enhance model reliability and accuracy.

| N-Ref | Mi-FR | Mi-FU | Ma-CR | CR | Mi-FR | Mi-FU | Ma-CR | CR | Mi-FR | Mi-FU | Ma-CR | CR |
|---|---|---|---|---|---|---|---|---|---|---|---|---|
| | | | | | | LLaMA2(7B) | | | | | | |
| | | | NQ | | | | PoPQA | | | | TriviaQA | |
| 2 | 46.99% | 21.05% | 11.47% | 75.71% | 35.39% | 28.75% | 7.87% | 75.81% | 43.45% | 23.75% | 9.24% | 74.95% |
| 3 | 50.70% | 18.18% | 13.67% | **76.33%** | 38.02% | 27.26% | 9.34% | **77.08%** | 46.70% | 20.90% | 11.20% | 75.25% |
| 4 | **51.87%** | 17.07% | **13.94%** | 75.77% | **43.82%** | 21.29% | **12.01%** | 76.88% | **50.60%** | 18.38% | **13.14%** | **77.17%** |
| | | | | | | LLaMA3.1(8B) | | | | | | |
| | | | NQ | | | | PoPQA | | | | TriviaQA | |
| 2 | 68.57% | 5.09% | 25.84% | 80.97% | 51.04% | 13.13% | 18.24% | 79.45% | 66.73% | 7.81% | 23.87% | 81.23% |
| 3 | 69.07% | 5.05% | 26.51% | **81.26%** | 51.81% | 13.09% | 18.04% | 79.38% | 67.41% | 7.20% | 24.14% | 81.48% |
| 4 | **69.80%** | 4.84% | **27.00%** | 81.20% | **60.56%** | 14.95% | **21.94%** | **82.77%** | **69.43%** | 6.36% | **26.75%** | **81.79%** |

Table 6: Results with different references for SREF. CR represents the consistency score of the correctly answered data.

**Prompts for SREF**

**Prompts for question generation $x_{pt}$:**
Rewrite the question '' in three different ways, ensuring that the answer remains '' for each version. Separate each version with a newline character.

**Example of $x_{pt}$:**
Rewrite the question 'Who got the first Nobel prize in physics?' in three different ways, ensuring that the answer remains 'Wilhelm Conrad Röntgen' for each version. Separate each version with a newline character.

**Prompts for multi-choice question $p$:**
Please select a single best answer for the multi-choice question. Please do not give anything other than the answer option. If you don't know the answer, choose the 'Unsure' option. /n References: '' /n Question: '' /n Answer:

**Example of $p$:**
Please select a single best answer for the multi-choice question. Please do not give anything other than the answer option. If you don't know the answer, choose the 'Unsure' option. /n References: Reference 1: *Who was awarded the inaugural Nobel Prize in Physics? A: The first Nobel Prize in Physics was awarded to Wilhelm Conrad Röntgen in 1901 for his discovery of X-ray*
Reference 2: *In what name was the first Nobel Prize in Physics bestowed in 1901? A: The first Nobel Prize in Physics was awarded in 1901 to Wilhelm Conrad Röntgen in recognition of ...*
Reference 3: *Who took home the first-ever Nobel Prize in Physics in 1901? A: Wilhelm Conrad Röntgen. Röntgen was a German physicist who discovered X-rays ...* /n
Question: Who got the first Nobel prize in physics? /n
Options:
A. Wilhelm Conrad R¨ontgen B. Unsure C. Einstein Albert D. Leonardo E. Marie Curie /n
Answer:

Figure 7: Prompts Example

**Prompts for Baselines**

**Prompts for RCI**

1) Please select a single best answer for the multi-choice question. Please do not give anything other than the answer option. If you don't know the answer, choose the unsure option. /n /n Question: /n Answer:

2) You selected . Now, evaluate your choice by comparing it to the other options. Explain why you think your choice is correct or why one of the other options might be better./n

3) Based on your evaluation, tell me your final Answer: /n

**Prompts for SRF**

1) Please select a single best answer for the multi-choice question. Please do not give anything other than the answer option. If you don't know the answer, choose the unsure option. /n /n Question: /n Answer:

2) You selected . /n Now, evaluate your choice by comparing it to the other options. Explain why you think your choice is correct or why one of the other options might be better./n

3) Based on your evaluation, tell me your final Answer:

**Prompts for SCK**

1) /n/nSentence: /n/nIs the sentence supported by the context above? Answer Yes or No./n/n Answer:

2) Based on the reference, answer the following questions:

3) Please select a single best answer for the multi-choice question. Please do not give anything other than the answer option. If you don't know the answer, choose the unsure option. /n /n Question: /n Answer:

**Prompts for SC**

1) Please select a single best answer for the multi-choice question. Please do not give anything other than the answer option. If you don't know the answer, choose the unsure option. /n /n Question: /n Answer:

*Sample a diverse set of answers.*

2) Aggregate the final answers.

Figure 8: Prompts for Baselines

