# OpenReview forum: "Entropy Reveals What You Know: An Entropy-Guided Method for Enhancing the Reliability of Large Language Models"
_ICLR.cc/2025/Conference — ICLR 2025 Conference Withdrawn Submission_

### Official Review · Reviewer_U4tH · 2024-10-19

**Soundness:** 3
**Presentation:** 3
**Contribution:** 3
**Rating:** 5
**Confidence:** 4

**Summary:**

In this paper, the authors propose a new method called **SREF** to enhance the reliability of LLMs and reduce hallucinations. The method involves rephrasing questions to introduce diverse responses based on the same knowledge, followed by concatenating these QA pairs as references for the final answer. Extensive experiments are conducted on several models and datasets, demonstrating that SREF achieves the best mean performance. The authors also correlate entropy with consistency and KL divergence, showing significant relationships between these metrics.

**Strengths:**

- This paper is the first to correlate using LLMs as references with entropy, thereby establishing a connection with consistency and KL divergence. The experimental results demonstrate a positive correlation between the decreased entropy by using LLMs as references and increased consistency and KL divergence.

- Comprehensive experiments are conducted from the perspectives of factuality and consistency across three datasets and twelve LLMs, providing strong evidence for the effectiveness of SREF.

- The paper is well-written and easy to follow.

**Weaknesses:**

- The statement, "With the influence of R, y will be 'Unsure' when R is inconsistenency with x, see Eq.7 for details," requires clarification. If R is contradictory, will the LLMs definitely provide an "Unsure" answer? More explicit reasoning is needed here.

- Although SREF performs well for small-scale models like Llama and Mistral, its performance on larger models such as GPT-4o is not as satisfactory, even leading to a decrease in performance. The reason for this should be clarified. Additionally, the variation in SREF's performance across different datasets raises questions about its generalizability.

- In Figure 3, some points that were originally marked as "Unsure" now appear inaccurate. Previous research has suggested that LLMs do not exhibit the same level of self-knowledge as humans. For genuinely unknown knowledge, LLMs may respond as if they are certain. In such cases, adding more references could lead to increased hallucinations. This implies that SREF can effectively address "known knowns" and "unknown unknowns," but may worsen performance in ambiguous situations, hindering its broader application.

- To better demonstrate why SREF diverges from the vanilla setting, a cross-dataset error analysis would be helpful. This would provide insight into specific areas where SREF succeeds or fails, allowing for a deeper understanding of its effectiveness.

**Questions:**

Typo: "With the influence of R, y will be 'Unsure' when R is inconsistent with x, see Eq.7 for details," "inconsistency" should be replaced with "inconsistent."

---

### Official Review · Reviewer_Xp1R · 2024-10-28

**Soundness:** 2
**Presentation:** 1
**Contribution:** 2
**Rating:** 3
**Confidence:** 3

**Summary:**

This paper reveal that large language models (LLMs) carry vast amounts of knowledge, but they face issues with factual consistency and truthfulness, leading to unreliable responses and application risks. To improve model reliability, the authors introduce SREF, an entropy-based self-reference approach, which enhances reliability by increasing consistency in answers to known facts and encouraging refusal to answer uncertain questions.

**Strengths:**

1. The method is simple and straightforward, utilizing question rephrasing sampling to generate relevant knowledge and perform reasoning to optimize the consistency of traditional self-correction. This approach is applicable to both open-source and closed-source models.
2. From the perspective of entropy, the method offers relevant theoretical support for their approach.
3. The experimental design is relatively comprehensive, validating the effectiveness of the SREF. The discussion is also interesting.

**Weaknesses:**

1. There are performance limitations associated with strong models. The vanilla setting even outperforms SREF on the NQ and TQ tasks. Furthermore, the averaging of results (Mean metric) across all models in Tables 1 and 2 is problematic, as it appears to downplay the performance decline observed in strong models.

2. In addition, compared to vanilla and traditional self-correction methods, the multi-round generation approach significantly increases the computational burden in terms of FLOPs. I encourage the authors to provide a comparison of FLOPs budgets between SREF and other self-correction techniques.

3. Self-correction and self-consistency methods are typically employed to optimize answers in reasoning tasks [1] [2]. However, evaluating only NQ, TQ, and HQ common sense QA dataset is notably limited. I recommend that the authors include reasoning datasets such as GSM8K and MATH to enhance the scalability and robustness of their method.


[1] S3c-Math: Spontaneous Step-level Self-correction Makes Large Language Models Better Mathematical Reasoners

[2] DotaMath: Decomposition of Thought with Code Assistance and Self-correction for Mathematical Reasoning

**Questions:**

Please refer to the weaknesses section.

1. Why does SREF fail on strong models, especially considering that strong models are recognized to have a broader self-knowledge that could enhance the generation of self-references? Please ask the authors to explain.

---

### Official Review · Reviewer_9jAM · 2024-10-30

**Soundness:** 2
**Presentation:** 1
**Contribution:** 1
**Rating:** 3
**Confidence:** 4

**Summary:**

The authors introduce an entropy-guided framework that leverages models' understanding of rephrased questions as relevant references, to improve answer consistency for known facts and encourage refusal of uncertain questions. Experiments on several LLMs show that SREF yields more reliable results for task performance and consistency.

**Strengths:**

The experiments cover 12 LLMs across 3 datasets, which is a good range.

**Weaknesses:**

The proposed method lacks novelty.  See related works such as  1) https://aclanthology.org/2023.findings-emnlp.1032/ 2) https://arxiv.org/pdf/2406.02543. 3) https://arxiv.org/pdf/2402.00367. Additionally, the motivation, especially in the introduction, isn’t clearly articulated. It’s unclear how limitations in intrinsic self-correction led to the decision to leverage the model's internal knowledge to create relevant references.


The experiments are inadequate. First, missing related convincing baselines: 1) https://aclanthology.org/2023.findings-emnlp.1032/
2) https://arxiv.org/pdf/2210.01296. Also, the evaluation of uncertainty is insufficient; a higher frequency of 'unsure' outputs does not necessarily indicate well-assessed uncertainty. A better evaluation would be to check if the model’s accuracy is higher when it doesn’t output ‘unsure.’

**Questions:**

1. Why rely on distractors and a multiple-choice setup to measure consistency? A more direct approach would be to directly measure the consistency of multiple generated answers.

2. Could you explain in detail how Equation 7 determines whether the model will be ‘unsure’ about a question?

3. The question in Figure 2 is itself ambiguous, requiring more than just factual knowledge. Could you provide examples where the question is unambiguous and SREF still shows improved performance?

---

### Note · Authors · 2024-11-21

I have read and agree with the venue's withdrawal policy on behalf of myself and my co-authors.